

# The internal seiche field in the changing South Aral Sea (2006-2013)

Elena Roget [1], Elizaveta Khimchenko [2], Francesc Forcat [1], Peter Zavialov [2]

[1] University of Girona, Catalonia, Spain.
[2] Shirshov Institute of Oceanology – RAS, Moscow, Russia.

*Correspondence to:* Elena Roget (elena.roget@udg.edu)

**Abstract.** Internal standing waves (seiches) in the South Aral Sea are studied for the first time. The study, based on numerical simulations and field data and, focuses on two different campaigns: the first in autumn 2006, when the stratification was weak, and the second in autumn 2013, when it was strong. During this time, the sea surface level decreased 3.2 m but differences in stratification depend only on the time when the campaigns were carried out because the maximum

density gradient decreases even though the sea continues to become saltier. In 2006 there was a mild prevailing northeasterly wind and in 2013 a mild easterly wind. The fundamental modes were identified as 36 h and 14 h, respectively. For both years we focused on the sub-inertial modes which were found to be either second or third vertical modes. In general, the vertical modes in 2013 are higher because of the strong stratification. For both years, it was found that the quasi homogeneous mixed deep upper layer could sustain internal waves under mild wind conditions. Regarding the horizontal

structure, in 2006 they are first and second modes and in 2013 second and third horizontal modes. The results suggest that, due to sea level variations, the neck connecting the Chernyshev Bay to the main body of the lake can become critical for the development a nodal line in that neck..

## 1 Introduction

The growing knowledge about surface and internal standing waves in lakes (seiches), beginning with the first documented

observations of surface oscillations in Lake Michigan in the seventeenth century, has been exhaustively reviewed by Hutter et al. (2011). The modern era of internal seiche study began with the Defant/Mortimer model (Mortimer, 1979) which can be extended to high vertical modes, that is, when a lake responds as a multilayer system. Nowadays, it is accepted that high vertical modes are often excited; however, observations of such modes were sparse until the end of the twentieth century (Heaps, 1961; LaZerte, 1980; Csanady, 1982). Nowadays, the importance of the internal wave field in redistributing wind

energy within lakes is well known (Wüest et al., 2000; Stocker and Imberger, 2003; Shimizu and Imberger, 2008) and different authors have focused on its impact on mixing (Stevens, 1999; Planella et al., 2011; Bernhardt and Kirillin, 2013), sediment resuspension (Bogucki and Redekopp, 2008) and sediment and phytoplankton transportation (Ji and Jin, 2006; Rolland et al,. 2013; Vidal et al., 2014). While studies and research into internal waves are now common for small and large lakes, for most of them, the characteristic internal seiche field has not been described in detail. Large lakes where internal





seiches have been studied, either theoretically or experimentally to varying degrees, include Lake Kinneret (Ou and Bennet, 1979; Antenucci et al., 2000), Lake Geneve (Lemmin et al., 2005), Lake Biwa (Saggio and Imberger, 1998) and Lake Michigan (Mortimer, 2004).

This paper presents the first study of the internal seiche field in the Aral Sea, which is located in Kazakhstan and Uzbekistan
(Central Asia). At present, it covers approximately 10 percent of its surface area in the 1960s and holds less than 10 percent of its former volume, now contained within two main water bodies: the North and South Aral seas (also known as the Small Sea and the Large Sea). In 2009, the shallow western lobe of the South Aral Sea dried up completely, and after partial replenishment, it has dried up again (UNEP-GEAS, 2014). However, what remains of the South Aral Sea, a shadow of its former self, is still a large lake about 150 km long and 25 km wide, on average.

The Aral Sea's desiccation has had an enormous human, ecological, and climatic impact (Micklin, 2007; Arashkevich et al., 2009; Zhitina, 2011; Rubinstein et al., 2014) and has become a paradigm of the large number of lakes all over the world whose water levels have dropped (Bai et al., 2011; Yildirim et al., 2011; Lauwaet et al., 2012; Tourian et al., 2015). The ongoing changes in the Aral Sea are being closely followed by the scientific community (Singh et al., 2012; Schettler et al., 2013; Shi et al., 2014). Since 2002, the evolution of its stratification have been documented (Zavialov, 2005, 2009, 2012),
the predominant circulation has been described (Izhitskiy et al., 2014a), and the role played by geomorphological processes associated with hydrodynamics in the evolution of a lake has been discussed (Roget et al., 2009). Regarding the internal waves, due to the lack of data, only very preliminary works exist. However, in 2013, during a field campaign which was not focused on the study of the internal waves, continuous temperature data within the water column were recorded for the first time, thus making it possible to assess the numerical results for the internal seiches using the Princeton Ocean Model (POM).
The POM, which is widely known, has been used to make marine forecasts for the U.S. Great Lakes (NOAA-GLERL, 2016), to study internal waves (Ueda et al., 2003; Munroe et al., 2005; Babu et al., 2011) and already applied to the Aral Sea (Roget et al., 2009; Izhitskiy et al., 2014a).

Here we present a numerical study of the internal seiches for the conditions in autumn 2006 and 2013, which is compared with the measured data. More precisely, we focus the analysis of the modes of oscillation below the inertial period in the
region, which is about 17 h, and larger than 3 h. This range has been established considering the length of the measured data series and the uncertainties in the simulation setup, as discussed in the Material and Methods section. Material and Methods is organized into different subsections presenting a description of the measurements taken which are relevant to this study, the numerical simulations run for the two campaigns, and the methods of analysis.

The Results section is also organized into subsections. In the first subsection, the spectral analysis of the field data is
compared with the numerical results predicted by the model at the same stations. In the following subsections, the modes





identified in the spectral analysis are studied all over the lake. In the Discussion section, different aspects concerning the horizontal and vertical structure of the internal seiches and how they differ between the two campaigns, as well as observations from other lakes, are commented. Lastly, the Conclusion is presented.

## 2 Materials and Methods

### 2.1 Site and measurements

The field data analyzed in this paper to support the numerical simulations were recorded during two field campaigns from 27–30 September 2006 and 29 October–3 November 2013, described in detail by Zavialov et al. (2008) and Izhitskiy et al. (2014b), respectively. Fig. 1a presents the shore lines of the South Aral in 2006 and 2013 together with the bathymetric levels for 2013, when the see surface level decreased 3.2 m. The shore line has mainly receded along the eastern shore, where the lake is shallower. Furthermore, the neck connecting the northern part of the South Aral Sea (Chernyshev Bay) to

the main body of the lake has narrowed considerably. The measuring stations in 2006 and 2013 are also indicated in Fig. 1a. The choice of sampling sites was limited due to difficult access to the sea and resulting logistical restrictions.

    In 2006, a Nortek Aquadopp acoustic Doppler current meter was deployed for 3 days at a depth of 39 m at station A2, in the deeper part of the lake. In 2013, bottom velocities were recorded with a Seahorse tilt current meter at a depth of 25 m at

station W1, close to the western shore. At the same station, a Star-Oddi thermistor chain was deployed with five temperature sensors located at depths of 15 m, 21 m, 22 m, 23 m and 24 m, which are not optimally located for this study but are useful to validate the numerical results. Other moorings deployed during the campaigns were too short and had to be disregarded. All the data series recorded in 2013 and analyzed here are from a two-day period. Wind speed was recorded with a portable automatic meteorological station installed on the western shore of the lake near station W1.

For both years, all data were recorded continuously at a sampling rate of 30 seconds, but for this study they have been averaged every 10 minutes. At the beginning of both campaigns, a CTD profile was taken at station A2 and water samples were collected at different depths. For each campaign, salinity values were obtained from samples processed in the laboratory using the standard dry residue method. Next, a linear regression was calculated for each year between the sample salinity values and the corresponding salinity obtained from the CTD data using the UNESCO formula. These regressions

were used to obtain a salinity profile from the CTD data before density profiles were calculated using the Aral Sea density formula proposed by Gertman and Zavialov (2011). The characteristic density profiles for both campaigns are presented in Fig. 1b, where the elongated horizontal dots on the 2013 profile indicate the depths at which the temperature sensors in the chain were placed. The stratification was weak in 2006 and strong in 2013, but this is merely circumstantial; it depends solely on the time when the campaigns were carried out. Consequently, the profiles in Fig. 1b are not representative of the





evolution of the stratification of the present hypersaline lake, which generally becomes saltier as the maximum density gradient decreases (Izhitskiy et al., 2014a).

## 2.2 Numerical simulations

Numerical simulations for 2006 and 2013 were performed using the POM, which is based on hydrostatic three-dimensional

primitive equations with the Mellor and Yamada closure scheme and terrain-following sigma coordinates (POM, 2016). The density formula for the current Aral Sea obtained by Gertman and Zavialov (2011) was included in the model. The Aral Sea bathymetry used has a 967 m NS and 538 m EW resolution (Roget et al., 2009). For the vertical coordinates, 12 equidistant sigma levels were considered in 2006 and 17 in 2013. The model has a free surface and a split time step. For the simulations presented here, the time steps for the external and internal modes were 10 s and 500 s, respectively, both of which fulfill the

Courant-Friedrichs-Lewy criterion. However, given the uncertainties introduced in the simulations by using equal initial stratification conditions (temperature and salinity) in all of the lake model grids, and considering the limitation of the measured data series for comparison, the output of the model was fixed for every hour. Wind stress on the surface was calculated as $\tau = \rho_a C_D V_{10}^2$ with the drag coefficient $C_D$, as defined by Hasselmann (1988). A similar approach was used for the bottom stress, where the drag coefficient was calculated to fit the velocities at the first grid point nearest the bottom

boundary with the logarithmic law of the wall taking a bottom roughness of 0.01 m into account. For all simulations, the heat flux at all boundaries was set to zero and no mass input or output was considered. Zero normal velocities were used as the lateral boundary conditions.

In 2006, the model was forced by hourly wind data downloaded from the coupled global NCEP Climate Forecast System Reanalysis. Data were interpolated from Gaussian to a regular geographic grid with a 0.31250º step and later to the grid

model itself. The model ran for 10 days, but the results were only analyzed after the fifth day, coinciding with the experimental field campaign. During this time, there was a mild northeasterly wind, the prevailing wind direction in the area. For the 2013 simulation, the model was forced over the first 5 days by a constant and spatially uniform easterly wind of 3 m/s, which coincides with the mean speed and the dominant direction during the campaign, and afterwards by the wind measured in situ.

## 25 2.3 Methods of analysis

The horizontal structure of the internal wave field is analyzed based on the surface level deviations from the equilibrium in the numerical results. This method is based on the consideration that internal oscillations can be traced by the corresponding oscillations on the lake surface with a characteristic amplitude, $\zeta_s$, which can be approximately related to the amplitude of the displacement of internal the density interface (halocline), $\zeta_h$, as $\zeta_s = ((\rho_h - \rho_e)/\rho_h)\zeta_h$, where $\rho_e$ and $\rho_h$ are the densities at

the surface (epilimnion) and bottom (hypolimnion) layers (Hellström, 1941). In our study, which considers the density




stratification during the 2006 and 2013 campaigns, the amplitudes of the internal waves at the halocline induced by the corresponding wind fields are, respectively, about 1000 and 100 times larger than at the surface.

To study the modes of the internal seiches identified in the spectral analysis, the surface elevation predicted by the model is band-pass filtered around the corresponding frequency (or period). When the oscillation is reproduced by the model, the
filtered surface elevation shows the horizontal structure of the internal seiche. However, if the model does not reproduce the mode, the filtered surface elevation does not present a coherent structure all over the lake. The method was initially proposed by Mortimer (1963) and has subsequently been used by different authors, including Caloi et al. (1986), Sirkes (1987), Lemin et al. (2005) and Forcat et al. (2011). Note that the periods of the internal seiches identified at the surface are in a range completely different from the periods of the surface seiches (Kirillin et al., 2015).

In this study the numerical results are compared with the spectra of the vertical position of the isotherms (in 2013) and of the bottom velocities (in 2006 and 2013) measured in the lake. Velocity components along and across the lake were computed considering its 14º tilt with respect to the north-south direction. Limitations resulting from the length of the field data forced us to focus this study on short sub-inertial modes.

For the 2013 campaign, the density and the thermal profiles presented the same structure so that vertical displacements of the
isopycnals (lines of equal density) were obtained from the position of the isotherms (lines of equal temperature), which were calculated by linear interpolation of the temperature values measured at fixed depths. To analyze the measured and numerical data, spectral, filter and correlation analyses were carried out with standard MATLAB functions.

The vertical structure of the identified modes of the internal seiches is analyzed from the numerical results by band-pass filtering the horizontal velocity at all grid depths at a fixed point around the corresponding frequency (period). If the
oscillation is excited, the velocity sign within each layer is the same. However, if the velocities are band-passed around a period which is not reproduced by the model, no coherent structure regarding the velocity sign (direction) is observed within the water column.

## 3 Results

### 3.1 Spectral analysis of the numerical and field data

Fig. 2a presents the spectra of the simulated surface elevation in 2013 at measuring station W1 and at the location (45.4ºN, 58.6ºE). The spectrum at station W1 shows a wide peak at around 14 h and peaks at 7 and 4 h at both locations. The 95% confidence interval shown in the figure indicates that the observed peaks are statistically significant. Fig. 2b presents the





equivalent spectra in 2006 at measuring station A2. As observed, there are two peaks at around 11 h and 7 h which cannot be completely separated and a third peak at 5 h. All the peaks are statistically significant.

Fig. 3 shows the spectral analysis of the field data recorded in 2013 based on the vertical displacements of the 6ºC, 7ºC and 10ºC isotherms (Fig. 3a) and on the bottom velocities (Fig. 3b). In both figures peaks of energy are observed at frequencies corresponding to periods of around 14 h, 7 h and 4 h, according to previous numerical results. As expected, the confidence intervals, displayed at the left side of the plots, are relatively large due to the short length of the experimental data series. However, they are still robust, particularly if we consider that the spectral analysis of two completely different measurements shows the same results, which are in also agreement with the numerical simulations.

Fig. 4 presents the spectra of the bottom velocity components along and across the main axis of the lake measured in 2006. As can be observed, the peaks of energy at around 11 h, 7 h and 5 h are also in accordance with the numerical spectral results and will be discussed below. The shape of the spectra at the lower frequencies in the component along the lakes is affected by the window of 128 points used to give more statistical relevance to the shorter modes.

In accordance with the results above, the oscillations of 14 h, 7 h and 4 h in 2013 and 11 h, 7 h and 5 h in 2006 are analyzed in the following sections, with consideration given to the outputs of the numerical simulations commented on earlier in the Material and Methods section.

### 3.2 The energy peak of 14 h in 2013

According to Mortimer's (1953) formula, the ratio of the period of the fundamental internal seiche for two different stratifications should be equal to the inverse ratio of the square root of their normalized density difference at the halocline. Given that the normalized density difference in 2006 and 2013 are, respectively, 2/1077 and 12/1088 (Fig. 1b), the fundamental mode in 2013 is expected to be about 2.4 times less than in 2006. Accordingly, and considering that the numerical simulations for 2006 by Forcat (2013) showed a fundamental mode of 36 h, it is reasonable to consider that the peak of around 14 h, observed in both the numerical and the experimental spectra in 2013 at station W1, corresponds to the fundamental mode, which is also the more easily excited mode.

Based on the simulations, the horizontal structure of the fundamental mode in 2006 can be observed from the by-pass filtered surface elevation around a frequency of $1/36$ $h^{-1}$. The corresponding plots from a reference time (t = 0 h) and after 18 h and 36 h , respectively, presented in Fig. 5, show a complete oscillation. As is observed in all three plots, the surface elevation shifts smoothly along the lake and there is a single line where the surface elevation is zero (nodal line) which is located about 45.40º N. To one side of the nodal line the surface level is above the equilibrium level (positive values) and to the other side it is below it. The direction of the vertical displacements changes from one plot to the next when the time shift is half the





oscillation period. A standing oscillation with a single nodal line is known as the horizontal first mode or the fundamental mode. In general, the number of nodal lines corresponds to the mode number (Hutter et al., 2011).

However, the 14 h oscillation observed in 2013 from the spectra of the field and numerical data at station W1 cannot be identified over the whole lake from the bypass filtered surface elevation as is the case for the oscillation of 36 h in 2006.

Note, however, from Fig. 2a, that the peak at around 14 h at station W1 shows two maxima, which may indicate two oscillations with similar periods but different structures that cannot be isolated from each other because the output of the model was fixed for every hour. In fact, in Fig. 2a, at the location [45.4º N, 58.6º E], which coincides with the nodal line of the 36 h mode in 2016 (Fig. 5), the peak at 14 h is not observed, meaning that it could correspond to the first horizontal mode. Furthermore, because of the limited length of the field data series, the resolution might not be high enough to

determine two close oscillations from the measured data, which also present a wide peak at around 14 h. All in all, it can be concluded that the 14 h peak observed in both numerical and field data contains energy of the fundamental mode even though, in 2013, its horizontal structure all over the lake could not be determined. The horizontal structure of the internal waves of all the other modes studied in this paper can be observed from the simulated surface elevation, as shown in the following subsections.

### 15 3.3 Structure of the modes of 7 h and 4 h in 2013

In Fig. 6, the surface elevations after being bypass-filtered around the periods of 7 h and 4 h (Fig. 6a and Fig. 6b, respectively) are represented for three different times covering the corresponding periods. In Fig. 6a the whole lake is observed to oscillate longitudinally with a period of 7 h (first and last plots coincide) presenting two nodal lines, indicating the excitation of a second horizontal mode. The first nodal line is located at around 45.9º N, at the entrance of Chernyshev

Bay (see Fig. 1) and the second, which has a crosswise character, extends from 44.98º N at the eastern shore to 45.38º N at the western shore.

The mode of 4 h in 2013 is also reproduced by the POM model, coexisting with the other oscillations. In Fig. 6b, the 4 h standing wave presents three nodal lines located at around 44.78º N, 45.78º N and 45.90º N, which indicates a third horizontal mode. Note that in this case the northern nodal line is also located at the entrance to Chernyshev Bay.

The vertical structure of these modes can be determined from the evolution in time of the filtered horizontal velocity along the lake around 7 h and 4 h at all depth grids of the water column at the deeper part of the lake presented in the panels (a) and (c) of Fig. 7. Despite the low values of the velocity after being filtered, in both cases, the velocity sign is the same at consecutive depths, changing only at two fixed depths that coincide with the interfaces between layers with different velocity directions. The location of these interfaces remains constant over time, showing the persistence of the mode. In Fig. 7a,

which corresponds to the 7 h mode, three horizontal lines are plotted at about 6 m, 16 m and 26 m deep where the horizontal



velocity is zero (nodal points), indicating interfaces between layers. Accordingly, for the mode of 7 h is therefore a third vertical mode and therefore the lake behaves like four different layers in relation to the direction of the velocity (Hutter et al., 2011).

Temperature data recorded at fixed depths within the water column in 2013 support the idea that the 7 h mode is a third

vertical mode. Considering the depths of the temperature sensors along the thermistor chain we were able to obtain the vertical oscillation of the isotherms of 11ºC and 6.8ºC. When comparing the temperature profile (Fig. 7b) with Fig. 7a it is observed that the isotherms of 11ºC and 6.8ºC are located in the upper and lower parts of the third layer which, because of the standing oscillation, should alternately approach and retreat and therefore move out of phase (Pérez-Losada et al., 2003). Unfortunately, the measuring station W1 was located relatively close to the horizontal nodal line (Fig. 6a), so vertical

displacements are relatively small. However, even in this case and considering the short length of the recorded data series, the correlation function between the 11ºC and 6.8ºC isotherms, presented in Fig. 8, shows a minimum when the time shift is zero, confirming that both isotherms oscillate out of phase. No temperature data were recorded within the other layers, but because the data available corroborates the location and the behavior of one of the predicted layers, and the period of the observed and numerical oscillations coincides, the excitation of the third vertical mode is indirectly corroborated. As for the

vertical structure of the mode of 4 h, Fig. 7c shows that it corresponds to a second vertical mode. That is, the lake behaves as a three-layer system and at two depths within the water column (at about 15 m and 25 m) the horizontal velocity is zero.

All in all, we can conclude that the standing oscillation of 7 h in 2013 corresponds to a second horizontal (two nodal lines) and a third vertical (four layers) mode and that that of 4 h corresponds to a third horizontal (three nodal lines) and a second vertical (three layers) mode.

**3.4 Structure of the modes of 11 h and 5 h in 2006**

Fig. 9a presents the surface elevation bypass filtered at around 11 h for three different times covering the oscillation. As can be observed, the single line where the surface elevation is zero (nodal line) is transversal to the main axis of the lake and located between 45.57º N and 45.40º N. Accordingly, the first horizontal mode oscillates longitudinally; that is, the surface elevations at the north and south sides of the nodal line have opposite signs. However, in the southern part of the lake, lines

of equal surface elevation are mainly longitudinal and therefore the wave presents a transversal structure. The transversal character of this oscillation at the southern part of the lake is in accordance with the results shown in Fig. 4, where a peak at 11 h is observed in the transverse component of the velocity measured at station W1 (Fig. 1a). Note that no nodal line developed at the entrance to Chernyshev Bay, unlike the case for all the analyzed modes in 2013.

The internal seiche of about 5 h in 2006 is also reproduced by the POM and the corresponding horizontal standing wave is

represented in Fig. 9b. In this case, the oscillation presents two nodal lines indicating the excitation of a second horizontal





mode. The northern nodal line is located between 45.92º N and 45.74º N, close to the entrance to Chernyshev Bay, and the second one about 45.1º N.

The simulated vertical structure of the modes and the corresponding filtered horizontal velocities are presented in Fig. 10. In both cases, the lake responds as if formed by three layers moving in opposite directions, consistent with the excitation of a second vertical mode. The interfaces between layers are indicated by the horizontal lines in the plots.

We can conclude that the standing oscillation of 11 h in 2006 corresponds to a first horizontal (one node), second vertical (three layers) mode and that that of 5 h in 2013 matches a second horizontal (two nodes), second vertical (three layers) mode.

### 3.5 The mode of 7 h in 2006

The mode of about 7 h observed in the spectra of the bottom velocity in 2006 can also be interpreted as a second horizontal mode according to the POM results. Fig. 11a represents its structure over the entire lake, based on the filtered surface elevation and presenting two nodal lines. On the other hand, in Fig. 11b, the corresponding vertical structure of the horizontal velocities shows three different layers. Accordingly, the mode of 7 h in 2006 is a second vertical second horizontal mode akin to the mode of 5 h but with a completely different horizontal structure.

## 4 Discussion

For both years, all the modes have been identified as longitudinal modes (that is, they oscillate along the main axis of the lake), even in the 2013 when there was a dominant easterly wind. In 2013, however, the longitudinal mode of 7 h has a crosswise nodal line encompassing 0.4º of latitude, and in 2006, when the dominant wind was northeasterly, the longitudinal mode of 11 h also presents a transversal structure at the southern part of the lake. Such behavior corresponds to what is observed in other lakes where excitement of longitudinal seiches with a transversal structure at the wider and deeper lobe is the standard response to an oblique or transversal wind forcing as in the case studied here (Roget et al., 1997).

In 2013, unlike in 2006, all the observed modes present a nodal line in the neck connecting Chernyshev Bay with the rest of the lake. For high horizontal modes, the development of nodal lines at locations where the transversal section is relatively small in comparison to the mean transversal area of the lake is well known (Roget et al., 1997; Imam et al., 2013). However, a decrease of 3.2 m in the surface level between 2006 and 2013 could have favored the development of nodal lines at the entrance to Chernyshev Bay in 2013, when the depth in the region was less than 5 m. Note that the development of a nodal line between Chernyshev Bay and the main body of the lake is not equivalent to the fact that Chernyshev Bay would have detached from the South Aral. While maximum horizontal velocities are found at the nodal lines, they are zero at the coast



(Hutter et al., 2011). Accordingly, the general structure of the oscillation and so the distribution of velocities affecting the exchange between different sub-basins and the mixing activity could be very different in one or other situation (Umlauf and Lemmin, 2005; Vidal et al., 2013; Guyennon et al., 2014).

Regarding the vertical structure of the studied internal seiches, only in 2013 was there a third vertical mode, all having been
second vertical in 2006. Higher vertical modes could have been expected in 2013 because of the relatively wide halocline and the strong stratification in comparison with 2006. A comparison of Fig. 1b with Fig. 7 leads to the conclusion that the halocline by itself behaves as a layer. On the other hand, the results from both years show that the quasi homogeneous upper mixed layer (0.12º/18 m in 2006 and 0.15º/18 m in 2013) can sustain internal waves, at least under the very light breeze conditions previous to and during the campaign. The importance of the stratification to the development of high vertical
modes has been discussed in detail by Pérez-Losada et al. (2003) and Imam et al. (2013), who also focus on the importance of irregular bathymetries. From our point of view, the relative complexity of field campaigns in large lakes prevents researchers from focusing on observations of the high vertical modes, which were no longer considered rare in small and medium stratified lakes and reservoirs once observations had begun to focus on them (Münnich et al., 1992; Vidal et al., 2007; Simpson et al., 2011).

Although the differences in the structure of the internal waves studied in 2006 and 2013 are mainly due to the development of higher horizontal modes and to stratification, the effect of the different wind forcing in 2006 and 2013 must also be taken into consideration. The structure of all possible internal seiches is known to depend on bathymetry and stratification, but seiches that are excited depend on the pattern of the wind (Fricker and Nepf, 2000). A number of authors have illustrated the importance of the wind pattern in determining the standing oscillations that become excited (Roget et al., 1997; Valerio et
al., 2012) and even in the shift of the internal waves from one mode to the other (Boehrer et al., 2000; Bastida et al., 2012).

As mentioned in the Introduction, comparisons with the results presented here are impossible because detailed studies of the high vertical modes in large lakes do not yet exist. Furthermore, even the formulas to approximate the period of the fundamental mode (Mortimer, 1953; Wüest and Farmer, 2003) are too complicated to apply to irregular or multi-basin lakes (Rueda and Schladow, 2002). Worth mentioning, however, is that Hutter et al. (1983) reported a second vertical first
horizontal mode of 12 h in Lake Lugano in Italy and Switzerland; Antenucci et al. (2000) found second and third vertical rotatory modes of about 12 h with differences of 1–2 h in Lake Kinneret in Israel; Prigo et al. (1991) reported a second vertical first horizontal mode of 2.5 h in Lake Champlain in the United States and Canada; Boehrer et al. (2000) found a second vertical mode larger than 6 h in the west of Lake Constance in Germany; and Valerio et al. (2012) reported a second vertical first horizontal mode of 56 h in Lake Iseo, Italy.






## 5 Conclusion

In this paper we have analyzed the internal seiches measured in the South Aral Sea in September 2006, when the lake was weakly stratified and a northeasterly wind was dominant, and again in October 2013, when the stratification was strong and there was a dominant easterly wind. Between 2006 and 2013, the sea surface level decreased 3.2 m. The study is based on
field data and numerical simulations using the Princeton Ocean Model. More precisely, we have focused on sub-inertial modes (bellow 17 h) larger than a 3 h period.

Our results show that the fundamental modes in 2006 and 2013 were of 36 h and 14 h, respectively. Further, in autumn 2006 there was a second vertical first horizontal mode of 11 h and two different second vertical second horizontal modes of 7 h and 5 h, while in autumn 2013, there was a third vertical second horizontal of 7 h and a second vertical third horizontal mode
of 4 h. Note that only in 2013 was there a third horizontal mode. All the studied modes are longitudinal, although the mode of 11 h in 2006 has a transversal structure at the southern part of the sea and the mode of 7 h in 2013 presents a crosswise nodal line.

This study points to the relevance of sea level variation for the excitation of higher horizontal modes. Shallower areas connecting different sub-basins can become critical for nodal line development, as is the case of the neck connecting the
northern part (Chernyshev Bay) with the main body of the South Aral. In this case, the drastic changes in the standing oscillations of horizontal velocities and vertical displacements could influence the distribution of energy mixing and phytoplankton exposure to sunlight.

On the other hand, the strong stratification during the 2013 campaign has favored high vertical modes. In both years, the quasi homogeneous upper mixed layer of 18 m was found to sustain internal waves under the light breeze conditions that
also favor high vertical modes.

*Acknowledgements*

The present work was carried out within the framework of the CLIMSEAS exchange project funded by the European Commission (Ref. FP7-IRSES-2009-247512). Field work was supported by Russian Science Foundation grant 14-50-00095. We are grateful to A.N. Serebryany for providing thermistor sensors for the field campaign and to V. Tischenko for
downloading and preparing the wind data reanalysis used for the numerical simulation. We also thank A.S. Izhitskiy for his help with questions concerning CTD and velocity data.



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





**Figures**

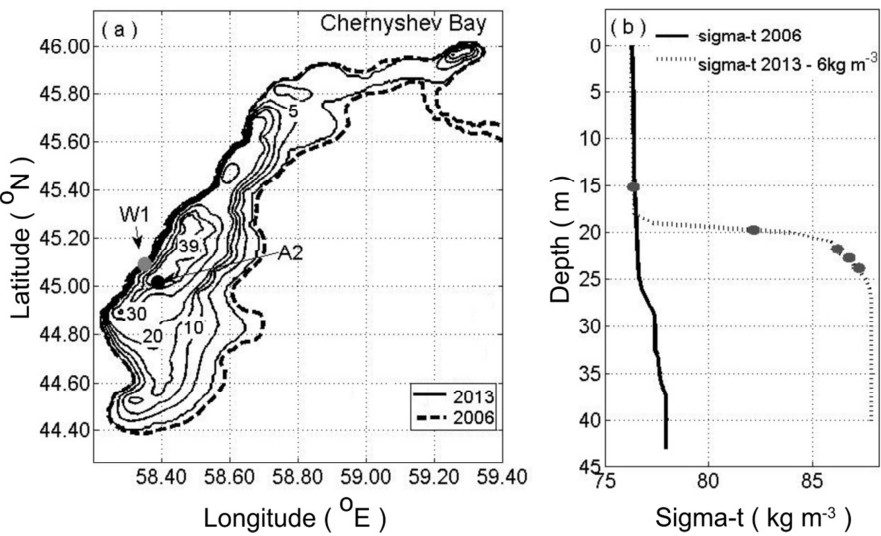

**Figure 1:** (a) Shore line of the South Aral Sea in 2006 and 2013 and bathymetric levels in 2013. Locations of the measurement stations (Station W1 in 2013 and Station A2 in 2006) are indicated. (b) Characteristic density profile during field expeditions in 2006 and 2013. Elongated dots on the 2013 profile show the location of the temperature sensors.

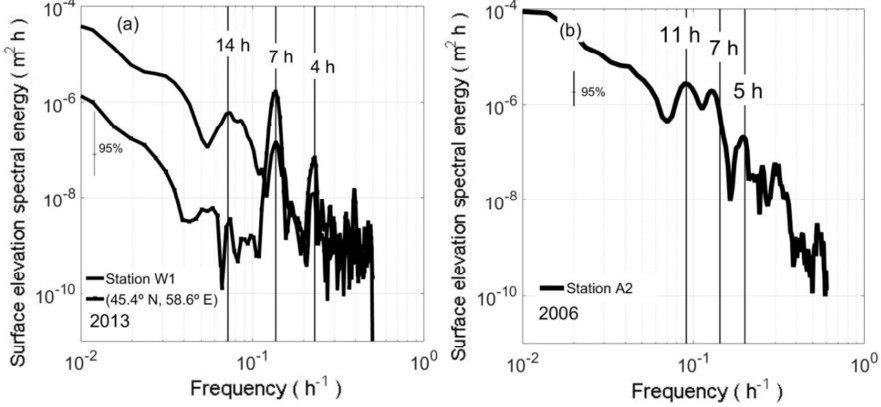

**Figure 2:** Spectra of the simulated surface level (a) in 2013 at station W1 and at (45.4° N, 58.6° E) and (b) in 2006 at station A2.




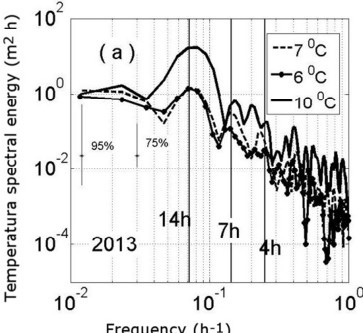 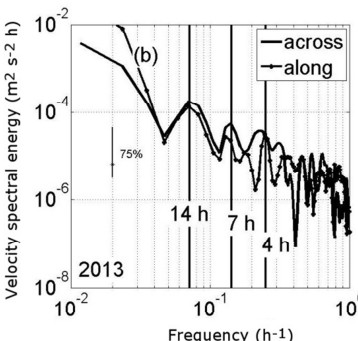

**Figure 3:** Spectra of (a) the vertical displacements of different isotherms and (b) bottom velocities measured at station W1 in 2013.

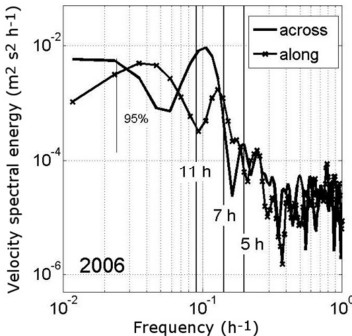

**Figure 4:** Spectra of bottom velocities recorded at station A2 in 2006.



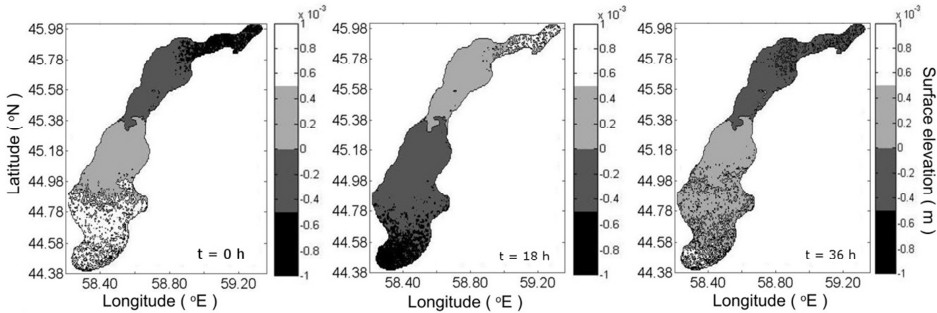

**Figure 5**: Band-pass filtered surface elevation for the oscillation of 36 h in 2006.

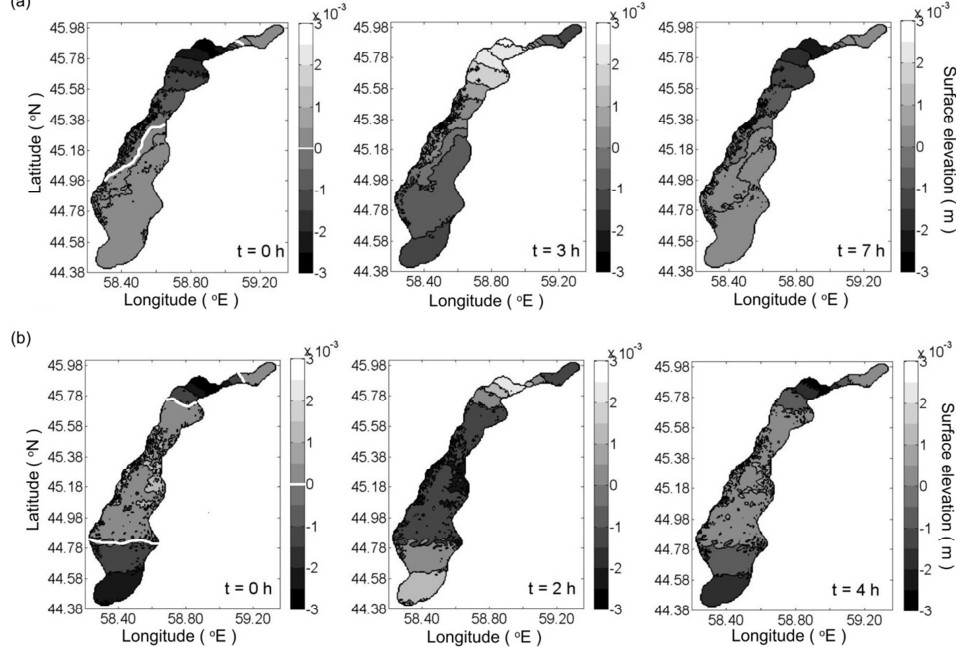

**Figure 6**: Band-pass filtered surface elevations for the oscillations of (a) 7 h and (b) 4 h in 2013.

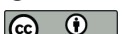


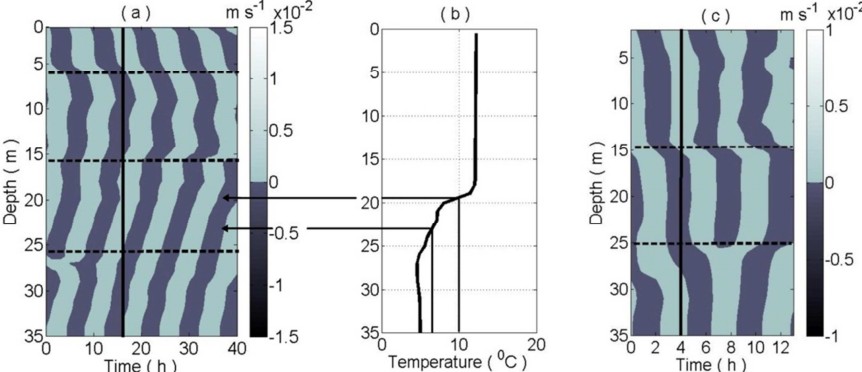

**Figure 7**: Filtered horizontal velocity along the main axis of the lake for the oscillations of (a) 7 h and (c) 4 h in 2013. (b) Measured temperature profile in 2013 showing the depths at 6.8° and 11ºC.

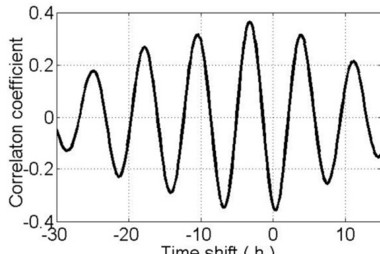

**Figure 8:** Correlation function between the vertical displacements of the isotherms of 11ºC and 6.8ºC measured at station W1 in 2013.





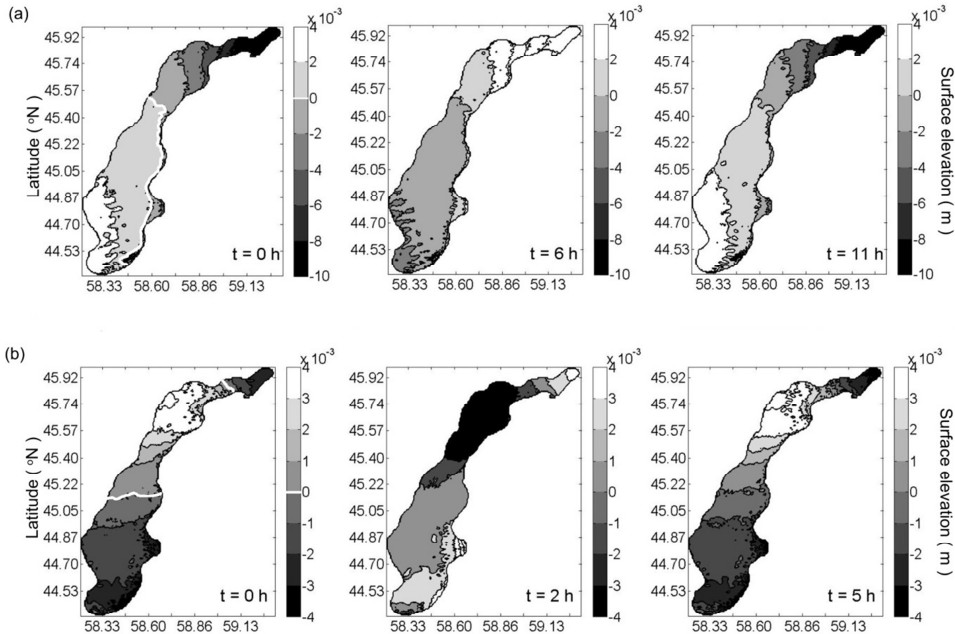

**Figure 9:** Band-pass filtered surface elevations for the oscillations of (a) 11 h and (b) 5 h in 2006.

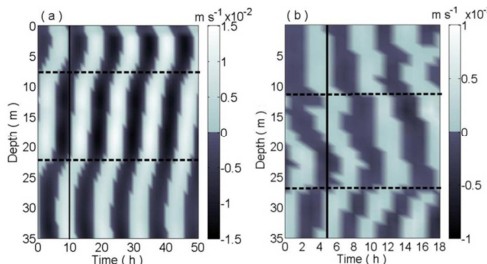

**Figure 10**: Filtered horizontal velocity along the main axis of the lake for the oscillation of (a) 11 h and (b) 5 h in 2006.




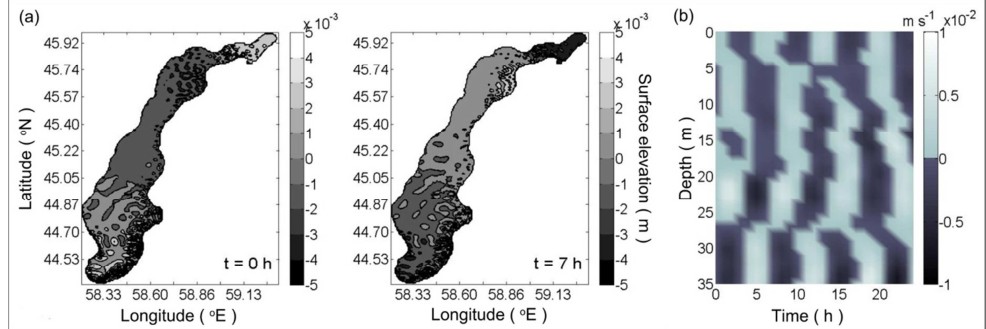

**Figure 11:** (a) Band-pass filtered surface elevation for the oscillation of 7 h  in 2006 and (b) the corresponding horizontal velocities along the lake.