# Peer review of "The internal seiche field in the changing South Aral Sea (2006-2013)"

_Hydrology and Earth System Sciences, 2016_

## Referee Comment (RC1) · K. Korotenko (Referee) · 2 Sep 2016

The paper is devoted to study of seiche oscillations on the base of velocity, temperature and salinity measurements and numerical simulations with a use of Princeton Ocean Model (POM). The study aim was to compare results obtained in 2006 and 2013 and understand how spectral structure of seiche has changed due to shallowing of the Aral Sea. The Aral Sea was once the world's fourth largest lake, slightly bigger than Lake Huron, and one of the world's most fertile regions. Today it is a dying sea, little more than a string of lakes scattered across central Asia east of the Caspian Sea. During the last 5-10 years the drying off of the Aral Sea, brought about noticeable changes in climate conditions has led to irreversible shrinking the Aral Sea. That is why any information, investigations that may help to reduce or stop this process is highly welcomed. In this connection, the reviewed paper presents a number of interesting

results on analysis of dynamic of the Aral Sea and its temporal shift due to changes in depth, the configuration of the sea coastline and salinity of the Sea during 8 years. At the same time, if instrumental part of the work is not in doubt, modeling with POM raises some questions: 1. It is said about mild wind as only source the model forcing, so that such wind hardly could be a cause of generation seiches during 10-day "spinning up" of the model, at least for energetic low frequencies as 17 and 14 h. Is it enough 10 days for spin-up? Has the control of quasi-stationarity of kinetic energy been performed? 2. It is of interest to show and compare mean circulation and turbulence level for two cases: for 2006 when stratification was weak and for 2013 when it was strong. 3. How the 18-m mixed layer was reproduced, in numerical simulation, particularly in 2013, when stratification was strong and winds were weak? 4. Was it good to use equidistant sigma-levels in such asymmetrical water body as the Aral Sea, i.e., the steep western coast and the gentle eastern one? A transition T, S and V to z-levels might have a problem of misinterpretation of results obtained. 5. Was the general mean circulation after spin-up correspond to that obtained in ADCP measurements? 6. Fig. 5. Why so high level of noise is in the water elevation in the southern part of the sea? All these questions are technical but required clarifying for understanding adequateness of field measurement results and those obtained with the POM. In a whole, the work is interesting and I endorse its publication in the Journal HESS.

K.A. Korotenko, PhD, P.P. Shirshov Institute of Oceanology, Moscow, Russia.

---

## Referee Comment (RC2) · Anonymous Referee #2 · 6 Sep 2016

The authors use two datasets from the largest residual basin of the former Aral Sea coupled with numerical modeling to analyse the internal wave weather in this recently formed natural water body. The basin-scale internal oscillations are known to be the major mediator of the energy transport to the bulk of the water column in enclosed stratified basins. In this regard, the study makes an important contribution into the fundamental understanding of the vertical energy transport in large strongly stratified lakes. The reported findings are especially relevant to dynamics of extreme aquatic environments, where strong vertical stratification is created by salinity gradients. The analysis is based on observational data separated by several years, allowing to trace the changes in the internal wave dynamics caused by continuous decrease of the water level and strengthening of the density stratification. The example of the Aral Sea is both unique and extendable on other saline lakes and seas subject to dessication/temporary

filling with freshwater/strong seasonal variations in external forcing. Taking into account the uniqueness of the recent Aral Sea development, the results reported in the study definitely deserve a wider dissemination and can eventually be published by HESS. In order to deliver the message to the target audience, the presentation style of the paper should be significantly improved.

**1 General comments**

- While not the most critical drawback, the grammar and style require improvement. The language should be checked, preferrably with the help of a native speaker, to ensure correct understanding of the results by the reader.
- The structure of the results presentation and the superficial discussion are my major critical points. Jumping there and back between results from 2013 and 2006, between simulations, observations, and results of a previous unpublished study (Forcat 2013) makes it extremely difficult to follow the authors' findings and ideas. After several readings, I suggest the major result reported here is the existing of standing waves of complex forms, in particular, the "high vertical modes" as the result of the specific multilayer vertical density structure of the present Aral Sea. This result should be addressed in Discussion in a more general context:
- Reasons for the differences in the internal waves filed between 2006 and 2013 (discussed only briefly in the present paper)
- Reasons for excitement of "high vertical modes" and consequences for the basinscale hydrological and biogeochemical regimes. (left practically unattended in the present discussion. What are specific features of the vertical density structure in the modern Aral Sea favoring the multilayer seiche modes? Do these oscillations change the transport of heat and mass in the Aral Sea, or is this only an odd

phenomenon? Even if an ultimate answer to these questions is impossible in frames of this paper, they still should be discussed in the context of the current knowledge).

• Outlook: comparison to observations from other seas/lakes. The three sentences comprising the last paragraph of the paper can be hardly considered as a serious discussion on this subject.

To make it comprehensible for the reader, the structure of the 'Results' section should be amended by:

- adding a subsection on the vertical density structure and external forcing in 2006 vs 2013 before the presentation of the spectral energy distribution in both years [now (3.1)]
- starting with data of 2006, or giving a reason why results of 2013 should be discussed first.
- presenting results on the 36h gravest mode from 2006 instead of referencing to a conference presentation.
- merging the subsections 3.4 and 3.5

**2 Specific comments**

2.1 Abstract

P1L8: 'decreased to' or 'decreased by' 3.2m?

- P1L17 Remove double period
- СЗ

**2.2 Introduction**

P1L21 Lakes were treated as multilayer systems before Mortimer (1979), e.g. by Heaps (1961)

P2L24-25 replace 'below the inertial period... and larger than 3 h' with 'shorter than inertial period and longer than 3 h'

2.3 Materials and Methods

P3L6-7 Replace 'from 27-30' with 'on 27-30'. Add 'on' to '29 October'

P3L14 'Seahorse' – there is no such instrument. It is called 'TCM-1 tilt current meter' manufactured by 'Lowell Instruments LLC'. Provide configuration, resolution and accuracies for all instruments used in the campaign.

P3L23 Replace 'dry residue' with 'dry rest'

P4L1-2 What do you mean with 'the maximum density gradient decreases'? Clarify

P4L20 Any justification for the 5-day spin-up of the model?

P5L14 replace 'thermal' with 'temperature'

P5L14-15 Are salinity profiles available for the period of campaigns? Are they evrtically homogeneous?

**2.4 Results**

P6L6 Replace 'according to' with 'in agreement with'

P6L10 Remove 'as can be observed'; replace 'accordance' with 'agreement'

P6L17 Add 's' to 'period'

P6L24 According to which simulations? Those of Forcat et al. (2013)? Present all relevant results in the paper.

P6L25-P7L1 Move the sentence to the figure legend.

P7L1-P7L2 Move the sentence to the introduction, or delete.

P7L16-17 Remove the sentence, it the figure legend repetition.

2.5 Conclusion

First paragraph: it is not a conclusion. Remove.

P11L3: see remark on P1L8

PL18: Explain how strong stratification can favor high vertical modes.

- 2.6 Literature
- P30: replace 'Ueda' with 'Rueda'

2.7 Figures

Fig. 2: add minor ticks

Figs. 7, 10: explain what the horizontal and vertical straight lines mean.

---

## Author Comment (AC1) · 22 Sep 2016

Answers to Referee #1

In response to comments 1, 2 and 5 of the reviewer, we want to clarify that the numerical simulations did not aim to reproduce the mean circulation of the lake and its interaction with the basin-scale wave field. Rather, it was our objective to complement the field data on the analysis of the structure of the seiches measured. In the revised version now we explain that the structure of the internal standing waves depends only on the bathymetry and on the stratification (P4L28-P5L1). So, the internal seiches can be analyzed without considering the mean circulation. In fact, some previous authors have based the analysis of the internal wave patterns on unforced models (Salvadé et al., 1988; Guyennon et al., 2014).

On the other hand, from all the possible standing waves, the internal seiches that are excited depend on the wind forcing, which is why we have forced the POM model with the wind prior to and during the campaigns. According to Sakai et al. (2011), wind forcing events persisting for some fraction of the wave period excite wave modes and this excitement can persist for several periods. For the purpose of this work, a 5-day spin-up period of the model was found to be enough for the standing waves to develop, although a longer spin-up period might be needed to accurately describe the circulation of the lake. Lorrai et al. (2011) also used a spin-up period of 5 days of the 3D hydrostatic Boussinesq model to study the seiching dynamics of a medium-size lake. Note that if the POM model had not reproduced the modes studied in this paper, the surface elevation filtered around the periods of the observed oscillations would not have presented a coherent structure throughout the lake as it is demonstrated in the Results section (P5 L3-L7)

Regarding comment 3, the model was initialized with the temperature and salinity profiles measured in the field, so the upper mixed layer was already introduced on the initialization. Some previous published studies, e.g. Lorrai et al. (2011) used the same initialization scheme. Some explanatory sentences have been added to the revised text (P4 L13-15).

For the analysis of the horizontal structure of the internal seiches, no transition from sigma levels to z-levels was required because the study is based only on the structure of the surface layer. Avoiding extrapolation between different vertical levels is, in fact, an advantage of the method (P5 L20). It is true, however, that for the analysis of the vertical structure of the waves, such a transition is made but only at a deep station in the center of the lake where no interpolation problems arise.

The higher noise in some of the filtered surface elevations at the southern part of the lake mentioned by the reviewer might be due to the fact that the total depth and transversal sections at this region are larger than in the northern part of the lake, making the vertical displacements at the surface smaller. Note also that the noise is higher in the fundamental mode in 2006 (Fig. 6a) which as discussed in the paper (P7 L15-16), might be damped.

We thank the reviewer for his comments which helped to clarify important aspects in the manuscript and improve it.

---

## Author Comment (AC2) · 22 Sep 2016

Answers to Referee #2

After reading the reviewer's comments, we realized that when we presented the results we were distracted by the historical development of the work and didn't present the results in a logical way. The 2006 data were analyzed prior to the 2013 campaign. At that time, we found that while the observational and numerical results matched well, the vertical structure of the waves could not be assessed based only on the bottom velocities. That's why we never tried to publish the results but we presented them at a conference (Forcat 2013). Only after the 2013 campaign, we realized that the data recorded within the water column could help to assess the vertical modes. Therefore we analyzed the 2013 data in the same way the 2006 data had been analyzed. Unfortunately, as our analysis of the 2013 data brought us back to the 2006 study, we presented the results starting from that year. As the 2006 results had not been published, we incorporated them in the paper, but we should not have mentioned Forcat (2013). It only confuses the reader.

After considering the reviewer's requirements, we have changed the structure of the Results and Discussion sections. The results are presented starting in 2006, and the plots have been modified accordingly. We have added a small introduction to the Results section that mentions the main characteristics of the stratification and the wind conditions which are described previously in the Materials and Methods section, because they are needed to better explain the numerical simulations and the methods of analysis prior to the Results section. The plot with the characteristic density profiles has been complemented with the corresponding temperature and salinity profiles.

The Discussion section has been organized according to the horizontal and the vertical structure of the internal waves. In it we comment on the features that favor the excitation of the different modes and the reasons for the differences observed between 2006 and 2013. Finally, although the fluxes related to the internal seiches cannot be assessed using these data, based in our results, they have been briefly discussed in the context of current knowledge.

As mentioned in the previous version of the manuscript, differences of morphometry and stratification make it difficult to compare the observed sub-inertial internal waves in the Aral Sea with those studied in other lakes. Therefore, after reading the reviewer's comment, we removed the last paragraph of the Discussion section and added the references given there to the Introduction section. More recent works have now been added to the Discussion section.

The reviewer's specific comments have also been addressed in the new version of the manuscript. In our opinion the reference to the Defant/Mortimer model does not contradict to works by early researchers treating the lake as multilayer systems after mentioning Mortimer (1979). Note that the work of Heaps (1961) had already been mentioned in the first manuscript after the reference to Mortimer's work.

We regret the reviewer's complaint about the language. The paper had been reviewed by a native speaker. Now it has been reviewed by another native reviewer who has more experience with scientific texts.

We acknowledge and appreciate the reviewer's constructive comments. We believe they have helped considerably to improve the manuscript.

---

## Author Comment (AC3) · 22 Sep 2016

The comment was uploaded in the form of a supplement:
http://www.hydrol-earth-syst-sci-discuss.net/hess-2016-331/hess-2016-331-AC3-supplement.pdf

---

## Author Response (AR1)

Answers to Referee #1

In response to **comments 1, 2 and 5** of the reviewer, we want to clarify that the numerical simulations did not aim to reproduce the mean circulation of the lake and its interaction with the basin-scale wave field. Rather, it was our objective to complement the field data on the analysis of the structure of the seiches measured. In the revised version now we explain that the structure of the internal standing waves depends only on the bathymetry and on the stratification. So, the internal seiches can be analyzed without considering the mean circulation. In fact, some previous authors have based the analysis of the internal wave patterns on unforced models (Salvadé et al., 1988; Guyennon et al., 2014) (commented on P4 L30 – P5 L2).

On the other hand, from all the possible standing waves, the internal seiches that are excited depend on the wind forcing, which is why we have forced the POM model with the wind prior to and during the campaigns. According to Sakai et al. (2011), wind forcing events persisting for some fraction of the wave period excite wave modes and this excitement can persist for several periods. For the purpose of this work, a 5-day spin-up period of the model was found to be enough for the standing waves to develop, although a longer spin-up period might be needed to accurately describe the circulation of the lake. Lorrai et al. (2011) also used a spin-up period of 5 days of the 3D hydrostatic Boussinesq model to study the seiching dynamics of a medium-size lake. Note that if the POM model had not reproduced the modes studied in this paper, the surface elevation filtered around the periods of the observed oscillations would not have presented a coherent structure throughout the lake as it is demonstrated in the Results section (P5 L3 –L10).

Regarding **comment 3**, the model was initialized with the temperature and salinity profiles measured in the field so the upper mixed layer was already introduced on the initialization. Some previous published studies, e.g. Lorrai et al. (2011) used the same initialization scheme. A clearer sentence has been added to the revised text (P4 L10).

For the analysis of the horizontal structure of the internal seiches, no transition from sigma levels to z-levels was required **(comment 4)** because the study is based only on the structure of the surface layer. Avoiding interpolation between different vertical levels is, in fact, an advantage of the method (P5 L19 – L24).  It is true, however, that for the analysis of the vertical structure of the waves, such a transition is made but only at a deep station in the center of the lake where no interpolation problems arise.

The noise in some of the filtered surface elevations in the southern part of the lake (mentioned by the reviewer in **comment 6**) is unfortunately highlighted by the contour levels chosen for a better visualization of the horizontal structure of the wave over the whole lake.

We acknowledge and appreciate the reviewer for his comments which helped to clarify important aspects in the manuscript and improve it as a result.

Answers to Referee #2

We regret the reviewer's complaint about the language. The paper had been reviewed by a native speaker. Now it has been reviewed by another native reviewer who has more experience with scientific texts.

After reading the reviewer's comments, we realized that when we presented the results we were distracted by the historical development of the work and didn't present the results in a logical way. The 2006 data were analyzed prior to the 2013 campaign. At that time, we found that while the observational and numerical results matched well, the vertical structure of the waves could not be assessed based only on the bottom velocities. That's why we never tried to publish the results but we presented them at a conference (Forcat 2013). After the 2013 campaign, we realized that the data recorded within the water column could help to assess the vertical modes and so we analyzed them. Unfortunately, we presented the results starting from 2013 and, because the 2006 results had not been published, we incorporated them in the paper, but we should not have mentioned Forcat (2013) which only confuses the reader.

After considering the reviewer's General comments, **we have changed the structure of the Results and Discussion sections.** The results are presented starting in 2006, and **the plots have been modified accordingly**. We have added a small introduction to the Results section that mentions the main characteristics of the stratification and the wind conditions which are described previously in the Materials and Methods section. The plot with the characteristic density profiles has been complemented with the corresponding temperature and salinity profiles. Results of the 36 h mode from 2006 are now presented (Fig. 6a)).

The **Discussion section** has been organized according to the horizontal and the vertical structure of the internal waves. In it we comment on the features that favor the excitation of the different modes and the reasons for the differences observed between 2006 and 2013 (P10 L11 – L15; P11 L10 – L23). Finally, although the fluxes related to the internal seiches cannot be assessed using these data, based on our results, a brief discussion of the relevance of the evolution of the internal seiche field for the Aral Sea and other shallow lakes in the current climate scenario has been added (P10 L15 – L21; P11 L23 – P12 L6; P12 L18 – L24).

As mentioned in the previous version of the manuscript, differences of morphometry and stratification make it difficult to compare the observed sub-inertial internal waves in the Aral Sea with those studied in other lakes. Therefore, after reading the reviewer's comment, we removed the last paragraph of the Discussion section and added the references given there to the Introduction section. More recent works have now been added to the Discussion section.

**The reviewer's specific comments have also been addressed** in the new version of the manuscript. In our opinion the reference to the Defant/Mortimer model does not contradict to works by early researchers treating the lake as multilayer systems after mentioning Mortimer (1979). Note that the work of Heaps (1961) had already been mentioned in the first manuscript after the reference to Mortimer's work.

We acknowledge and appreciate the reviewer's constructive comments. We believe they have helped considerably to improve the manuscript.